# The limits of near field immersion microwave microscopy evaluated by imaging bilayer graphene moiré patterns

Douglas A. A. Ohlberg [1], Diego Tami [1,2], Andreij C. Gadelha [3], Eliel G. S. Neto [4], Fabiano C. Santana[3], Daniel Miranda [3], Wellington Avelino[2], Kenji Watanabe [5], Takashi Taniguchi [5], Leonardo C. Campos [3], Jhonattan C. Ramirez[2,6], Cássio Gonçalves do Rego[2,6], Ado Jorio[2,3,7] & Gilberto Medeiros-Ribeiro [2,8 ✉]

Near field scanning Microwave Impedance Microscopy can resolve structures as small as 1 nm using radiation with wavelengths of 0.1 m. Combining liquid immersion microscopy concepts with exquisite force control exerted on nanoscale water menisci, concentration of electromagnetic fields in nanometer-size regions was achieved. As a test material we use twisted bilayer graphene, because it provides a sample where the modulation of the moiré superstructure pattern can be systematically tuned from Ångstroms up to tens of nanometers. Here we demonstrate that a probe-to-pattern resolution of $10^8$ can be obtained by analyzing and adjusting the tip-sample distance influence on the dynamics of water meniscus formation and stability.

[1] Microscopy Center, Universidade Federal de Minas Gerais, Belo Horizonte, MG, Brazil. [2] Electrical Engineering Graduate Program, Universidade Federal de Minas Gerais, Belo Horizonte, MG, Brasil. [3] Physics Department, Universidade Federal de Minas Gerais, Belo Horizonte, MG, Brazil. [4] Instituto de Física, Universidade Federal da Bahia, Salvador, BA, Brazil. [5] National Institute for Materials Science (NIMS), Tsukuba-city, Ibaraki, Japan. [6] Department of Electronic Engineering, School of Engineering, Universidade Federal de Minas Gerais, Belo Horizonte, MG, Brazil. [7] Technology Innovation Graduate Program, Universidade Federal de Minas Gerais, Belo Horizonte, MG, Brazil. [8] Computer Science Department, Universidade Federal de Minas Gerais, Belo Horizonte, MG, Brazil. ✉email: gilberto@dcc.ufmg.br

Liquid immersion microscopy has its roots in observations made by Hooke in 1679[1] that images would improve in clearness and brightness upon spreading fluids onto the surface of a sample and gently elevating it until the liquid touched his microscope lens. In addition, the adhesion of liquid to the lens was so robust and firm that the liquid remained attached, even as the investigated sample was moved about the field of view. This vivid description of meniscus formation and usage was subsequently expanded in 1813 with Brewster's concept of the oil immersion lens[2]. Later, in 1855, Amici improved upon several construction aspects, concerned primarily with diminishing the loss of light in high-power microscopes by opting for water as the immersion liquid[2]. Ensuing developments that further addressed the issues of light loss and improvement of the magnification power of lenses consolidated the recognition of immersion lens microscopy as a well-established technique.

Albeit remarkable, all these developments are diffraction-limited, defined by Abbe's resolution limit of $d = \lambda/2n\sin\theta$, with $\lambda$ as the radiation wavelength, $n$ the refractive index, $n\sin\theta$, as the numerical aperture[3]. The proposal of scanning aperture imaging by Synge[4] for near field imaging was put into practice in 1972 by Ash[5], improving magnification beyond the Abbe limit with a figure of merit of $\lambda/d$ of 60. Molecular and atomic imaging required the development of scanning probe microscopies. Scattering-type near-field microscopy[6] and pico-cavity tip-enhanced Raman spectroscopy imaging later demonstrated that visible and infrared radiation[7] can surpass this limit by using scanning probe tips to access the near-field regime in an apertureless mode. The focus on near field regime has enabled a tremendous advance in microscopy, deserving a detailed and fair review that falls outside the scope of the present letter, as it would encompass implementations with different wavelengths, construction details, and application fields. In the microwave regimen, there are interesting opportunities to be explored, as the field is at the crossroads of optics and electronics.

The scanning microwave impedance microscope (sMIM) is one of the latest additions to the family of scanning probe microscopes. Commercially available[8] tools can now be used to retrofit existing equipment, and exciting results in multiple applications have been published[9–12] describing exquisite spatial detail and vector analysis of the microwave reflected signal at each pixel. A 3 GHz microwave signal is coupled to an Atomic Force Microscope (AFM) probe tip that works as a waveguide and performs as an apertureless near-field microscope[9]. A key differentiating aspect of sMIM is that, unlike Scanning Tunneling Microscopy (STM), its ability to image nano-scale modulations in the electronic and dielectric properties of complex structures is not restricted to conductive samples, but also allows imaging of insulating dielectrics as well. The capacitance signal conveys dielectric, geometric, and quantum information. Previously, Seabron[10,11] employed sMIM to assess the quantum capacitance of carbon nanotubes. Capacitance spectroscopy is a technique historically employed to map the electronic and quantum properties of quantum dots[13,14] and two-dimensional (2D) quantum systems[15]. In these systems, a dielectric layer is mandatory for a proper adjustment of the chemical potential between the probe electrode (gate, tip) and the system of interest (quantum structure). For the case of sMIM, Seabron[10] posited that to improve the spectroscopic resolution, a high permittivity capping layer would be essential to better couple the tip to the sample. This coupling layer can be modeled as a series capacitor, and increasing its capacitance maximizes the coupling to the quantum system. Later, it was noted[11] that the best coupling would be realized by adventitious water found on surfaces that spontaneously formed a meniscus. With a relative permittivity $\varepsilon_r = \varepsilon/\varepsilon_0 \approx 80$ ($\varepsilon$ and $\varepsilon_0$ as the absolute and vacuum permittivities), and

a refractive index $n$ of $\approx 9$ at 3 GHz frequencies, the effect of water must be included in any modeling of sMIM experiments at ambient conditions.

Twisted Bilayer Graphene (TBG) systems offer an extraordinary opportunity to create two-dimensional superlattices of varying periodicity in a conceptually simple strategy of adjusting the twist angle $\theta$ between the two graphene layers. The search for systems producing two-dimensional modulations of periodic potentials in appropriate dimensions has seen a variety of implementations over the years, with examples ranging from antidot lattices[16] and top-gate modulation[17] in 2D electron gases of III-V heterostructures in the 1990s to more recent and exciting TBG embodiments[18]. The possibility offered by van der Wall heterostructures such as graphene to explore the potential modulation parameters in a more detailed fashion presented surprising opportunities that went beyond metal-insulator transition and Wigner crystallization[17] when observation of additional electron-correlation physics such as superconductivity was reported in TBG with a magical angle of $\approx 1.1°$[19]. Tools that can expeditiously analyze and provide answers on the electronic structure, preferably at ambient conditions, are currently being pursued[12,20].

Here, we report sMIM results with $\approx 1$ nm spatial resolution performed on TBG systems of varying twist angles along with theoretical modeling which demonstrates how water menisci can concentrate electromagnetic fields within small regions. The conditions for meniscus nucleation and stability are also discussed.

## Results

**Microwave microscopy data**. Figure 1 shows a series of sMIM scans over a set of TBG systems with twist angles $\theta$ of **a** 0.21°, **b** 0.86°, **c** 0.93°, **d** 1.37°, **e** 4.54°, and **f** 6.7°. The experimental setup and imaging conditions are depicted in the "Methods section" and supplementary Fig. 1. These samples were characterized by Raman spectroscopy (supplementary Fig. 2), Tip Enhanced Raman Spectroscopy microscopy[21], and Ultra High Vacuum-STM (supplementary Fig. 3), to independently verify the bilayer locations and confirm the observed moiré superlattices that arise in TBG systems. When TBGs are deposited on atomically flat substrates, surface topography contributions to the reflected microwave signal are conveniently eliminated, leaving the underlying electronic and dielectric structure components in the admittance intact.

Figure 1 demonstrates the ability of sMIM to observe the solitonic structures that arise in the atomically reconstructed TBGs prepared with twist angles smaller than 1.1°[20] (Fig. 1a) and the change in periodicity as we move towards angles larger than 1.1°, beyond which, bilayers no longer atomically reconstruct and remain rigid with respect to each other (Fig. 1b through f).

The false-color scale is keyed to the intensity of the real part of the reflected microwave signal, i.e., the conductance component. All the data shown here are non-filtered, and the only image processing performed was background removal and color range adjustment. In the upper right corner of each image, a Fourier Transform (FT) of the data is displayed, showing diffraction spots corresponding to the periodic modulation of the electronic properties due to the moiré two-dimensional superlattice. The sequence spans a wide range of periods, culminating in a 6.7° twist angle and a period $1/f$ of 2.1 nm for the sample in **f**.

The Nyquist frequency $f_c$, defined by the highest frequency that can be inferred from a signal requires a spatial resolution of at least $2f_c$[22]. Thus, our resolution is better than 1.05 nm ($1/2f$), despite the fact that the tip radius is 50 times bigger. Considering the microwave radiation wavelength at $\approx 3$ GHz of 0.1 m, our figure of merit is $10^8$. The resolution at this wavelength under these experimental conditions requires a deeper investigation.

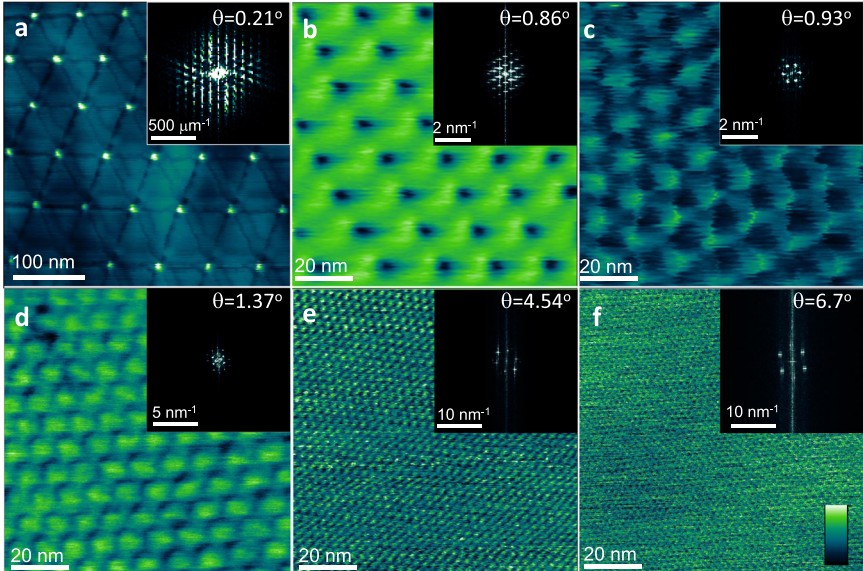

**Fig. 1 sMIM scans. a** 400 × 400 nm conductance image of TBG:hBN:glass, with contrast arising from the juxtaposing of two graphene layers with $\theta = 0.21°$, and the strain soliton domain wall arising from surface reconstruction clearly resolved[20]. The false color is keyed to the intensity of the reflected signal, white being the highest, i.e., higher conductance. The observed pattern is consistent with a reconstructed structure. The inset shows the corresponding Fourier Transform (FT). **b**–**f** 100 × 100 nm scans exhibiting a wide range of angles and corresponding moiré patterns. The systems examined are TBG:hBN: glass (**a**, **b**, **f**), TBG:SiO$_2$:Si **c**, and TBG:glass **d**. The color scale is shown in **f**. The full scale for images shown in **a** through **e** are of the order of hundreds of mV, whereas **f** is 30 mV).

**Tip-Surface Interaction**. Examining the tip-surface approach curves tracing cantilever deflection and the associated capacitance signal allows us to assess tip-surface mechanical and electrical coupling. Figure 2a schematically represents the meniscus model and the key parameters employed in the analysis. The observed capacitance and force behaviors observed during tip approach and retraction with respect to a TBG:hexagonal Boron Nitride (hBN): Glass stack are shown in solid black and red dotted lines in Fig. 2b. The directions of approach and retraction on the capacitance data are indicated by arrows. At about ~10 nm sample-substrate distance, the tip experiences capillarity attraction (~7 nN force), and the capacitance jumps, which can be explained by an additional contribution of the water meniscus capacitance. Following the onset of the tip deflection due to the attractive force of the meniscus and onward, the $x$-axis of the force/capacitance traces no longer represent tip-surface distance but rather a z-piezo displacement because of the cantilever elastic deformation towards the surface.

As a first approximation, the data are fit to an analytical model for the capacitance between a tip and a surface[23], described by Eq. (1) (orange line with solid circles):

$$C_{meas} = C_{stray} + 2\pi\varepsilon_0 R \ln\left[1 + R(1 - \sin\theta_0)/z\right] \quad (1)$$

with $C_{stray}$ as the stray capacitance, $\varepsilon_0$ the vacuum permittivity, $R$ as the tip radius, $\theta_0$ the aperture angle, chosen to be about 10°[23], and $z$ the tip height. The data and fit are plotted as $\Delta C = C(z) - C$ (1 μm), and multiplied by a normalizing constant. $C(1 \mu m)$ encompasses $C_{stray}$ and the capacitance between the tip and surface at $z = 1 \mu m$. The tip radius $R$ was kept fixed at 50 nm, its nominal value. The agreement between data and analytical model captures the capacitance dependence on $z$ from 1 μm to about 50 nm from the surface.

**Numerical modeling**. Since this analytic model is a first-order approximation of our experiment, we further modeled the system with a Finite Element Method[24] using the COMSOL$^{TM}$ Multiphysics simulation tool (see supplementary Figs. 4, 5, and relevant discussion), in order to incorporate the effect of meniscus formation and structures with more complex geometries and electronic properties and their corresponding effects on the reflected microwaves. The reflected microwave signal is a complex function that depends on substrate admittance with the sample conductivity and permittivity inextricably connected. The real and imaginary parts are related to the system conductance and capacitance.

Water is ubiquitous and frequently considered an unwanted nuisance that complicates nano-scale phenomena, but for sMIM experiments at 3 GHz its effects cannot be neglected as pointed out previously. AFM embodies one of the most convenient tools to probe capillarity at the nanoscale level. Experiments covering meniscus nucleation[25] and meniscus stiffness[26] illustrate the level of control and understanding that has been achieved of the tip-water meniscus-surface system. This exquisite control can be used as a resource to harness the meniscus geometry, with humidity, temperature, tip velocity, and tip-pulling force as the chief parameters.

The FEM model from which the capacitance dependence on $z$ is derived is shown as a black dashed line for a system with adventitious water of 1 nm on the tip and surface, and no water meniscus. The inclusion of menisci of radii $a$ of 3 and 6 nm adds an additional contribution to the capacitance, shown in solid light and dark blue solid circles. The top $x$-axis represents $t_{meniscus}$, the meniscus thickness for each solid circle. In the literature, the proposed values for meniscus thickness $t_{meniscus}$ are of the order of 0.2 nm in close proximity[27] to 5 nm at snap-off[28], depending on the ambient temperature and humidity. The smallest radii $a$ ranged from 1.3 nm to 2.6 nm[25,28]. Within the simplifying assumptions for the proposed geometry, amount of water on the surfaces, dynamics of menisci formation, and capacitive forces[23] that may pull the tip closer, the capacitance behavior derived from the simulated FEM model captures the essence of the tip approach and meniscus formation and closely matches the experimental data and analytical model.

**Meniscus dynamics and impact on imaging**. The dynamic aspect of meniscus formation[25], which for tip approach occurs

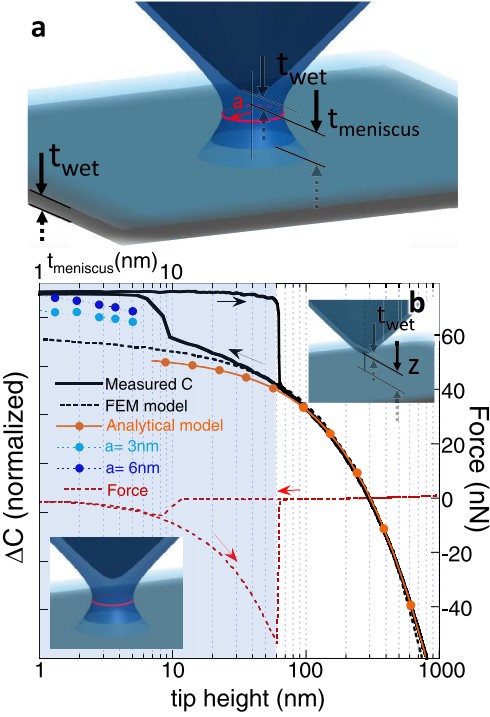

**Fig. 2 Meniscus model. a** Schematic diagram depicting the model parameters: $a$ is the meniscus radius. $t_{wet} = 1\,nm$, is the native water layer existent in all surfaces for typical scanning conditions. $t_{meniscus}$ is the meniscus thickness, measured from the water surface on the substrate to the water surface on the tip. Therefore, the distance between the tip to the surface is $t_{total} = 2t_{wet} + t_{meniscus}$. **b** Capacitance and tip deflection force data during tip approach in solid black and dashed red lines. Analytical and FEM models describing the tip-surface capacitance (orange line with solid circles and dashed black lines). The blue dashed lines and solid circles represent FEM model results for the tip, surface, and water meniscus system, with meniscus thickness $t_{meniscus}$ (upper $x$-axis) ranging from 1 to 6 nm, with radii $a$ of 3 nm and 6 nm (light and dark blue). For the tip approach cycle, the lower $x$-axis represents the tip-surface distance (excluding the $2t_{wet}$ contribution of the surface water layers) up until tip pull by the water meniscus until around 8 nm after which tip deflection is dominated by capillary forces and water meniscus formation. For the remaining travel of tip approach, and for the tip retraction cycle, the $x$-axis represents piezo-displacement, until the tip snaps off, at about 60 nm. The shaded blue represents the locus of sMIM operation with the presence of a water meniscus during tip retraction, i.e., an attractive mode where we conducted all sMIM experiments.

within the order of a few ms, impacts not only the force curves during the approach, but also scanning, generally speaking. In fact, for the scan rates employed (1 μm/s for Fig. 1c) the estimated meniscus radius is 2 nm[25]. As an additional test to corroborate the presence of a meniscus, we performed experiments in the so-called nap mode which basically is a set of two consecutive line scans, one in non-contact and the second at a pre-defined lift height (here we control the lift to sit below the tip snap off, and as such, during the lift, the meniscus is subject to a normal force of about 40–50 nN since the spring constants of the cantilevers used are typically 1 N/m). We were able to continue imaging at lift heights of 50 nm piezo displacement with a minor capacitance drop (~0.01ΔC) and sustained meniscus presence. In order to test the meniscus stability and nucleation dynamics, we ran a final scan at 2 μm/s, and a 300 nm lift (or 300 nN normal force onto the meniscus). For the first few tens of scan lines the meniscus under lift was stable and imaging, possible. After the initial scan

lines, it ruptured, at which point imaging of the moiré pattern was visible in close proximity, but no longer possible in lift mode, and the capacitance dropped at the lift scan (see supplementary Figs. 7, 8, and relevant discussion in the supplementary note 6: nap experiments).

The concentrating effect of the meniscus on the microwave fields can be visualized through FEM modeling. In Fig. 3 we examine the field distribution in the vicinity of and inside the water meniscus at $t_{meniscus} = 1$ nm. The FEM calculated distribution of $\mathbf{D} \equiv \varepsilon_0 \mathbf{E} + \mathbf{P}$ over the entire system and its evolution during tip approach can be seen in the supplementary movies 1 and 2 (with and without meniscus formation, respectively). The simulation results are shown in Fig. 3 exhibit the configurations of tip-substrate without meniscus Fig. 3a (wide view), b (detailed view) and with meniscus Fig. 3c (wide view), d (detailed view), on top of a TBG layer. The majority of the displacement field is localized at the apex of the tip upon meniscus formation, but noteworthy is the fact that it concentrates at the water:TBG interface, as seen in the **D** profiles (white lines) in Fig. 3b,d. From a microwave perspective, the meniscus is an iris that upon nucleation, control, and operation in the attractive mode allows for field concentration. Thus, menisci can be used to augment near-field resolving power. The Nyquist limit derived from the Fourier analysis of the moiré patterns represents an upper bound on resolving power and is consistent with the FEM modeling.

Near field and immersion optics at microwave frequencies create opportunities worth exploring. One convenient aspect of sMIM is the lack of externally coupled optical apparatus, enabling connectorized tools for easy deployment. Further possibilities can be envisioned for near-field immersion microscopy. The water layer requirement may for instance allow examination of biological samples, using single-layer talc or hBN sheets as cover-slips for adequate microwave transmission. The ability to implement spatially resolved capacitance spectroscopy by means of DC biasing schemes is an exciting prospect, becoming an invaluable tool for van der Walls heterostructures and band-gap engineering. An often explored resource of scanning probe microscopy is nanolithography. Yet, for the majority of the tools employed in nanolithography, the embodiments are normally implemented in an open-loop fashion, allowing only post-mortem inspection. With the reflected microwave signal, one can close the loop and monitor the complex impedance of the region of interest, while performing the patterning[29]. An immediate implementation in the already vast field of dip-pen nanolithography[30] would envision tracking both the real and imaginary parts of the reflected microwave signal to enable real-time tracking of minute quantities of dispensed materials, each with its impedance signature.

## Methods

**Preparation of TBG and survey protocol.** The twisted bilayer samples analyzed were prepared using a technique we have developed that is a variation of conventional, dry transfer, tear-and-stack methods[21,31]. Unlike other procedures, which either completely encapsulate graphene bilayers within a top and bottom layer of h-BN flakes or a bottom layer of h-BN flake and a top layer of polymer that often requires removal in subsequent solvent soaks and sample bakes, our dry transfer procedure fabricates simple, extremely clean, and unencapsulated TBGs free of polymers that can introduce undesired contaminants. The procedure relies on a special stamp design consisting of a truncated, polymer pyramid fabricated on a handle substrate that is not only capable of performing tear-and-stack operations on the initial graphene but also allows subsequent detachment of the bilayer onto a variety of support substrates. The substrate supports included simple glass cover-slips with and without h-BN coating layers that were used for tip-enhanced Raman spectroscopy (TERS) analysis (published elsewhere[21]), mica coupons coated with atomically flat gold for STM analysis, and silicon coupons coated with a 275 nm oxide. After transfer to a respective substrate, a WITec Alpha 300 SAR confocal Raman Microscope was used for Raman spectroscopy and spatial mapping. These measurements were typically performed using a 633 nm laser, power of 5 mW, and

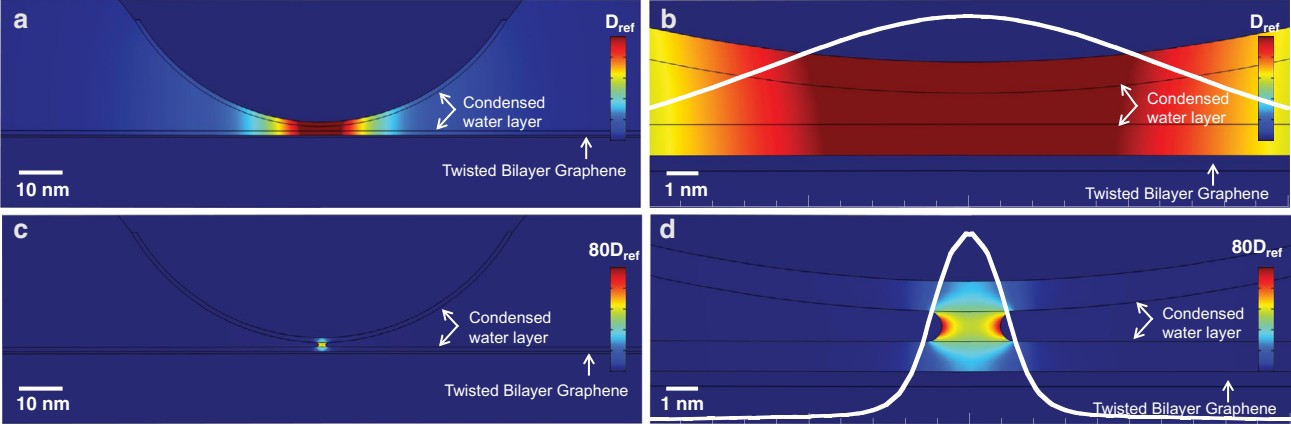

**Fig. 3 Electric displacement field distribution in the tip/sample simulated structure.** $D_{Ref} = 2.5 \times 10^{-3} C/m^2$ **a** Tip/sample simulated system for Graphene:hBN:Glass substrate at a distance $z=1$ nm without meniscus, and a 1 nm layer of water on both surfaces. Image total lateral size = 100 nm. **b** Close-up of the structure detailed in **a**. Image total lateral size = 10 nm. **c** Simulation for the same structure in **a**, with the additional implementation of a water meniscus. Image total lateral size = 100 nm. The color range has been expanded by $D_{fullscale} = 80 D_{Ref}$ in order to permit the visualization of the increased density in **D**. **d** Close-up of the structure detailed in **c**, also with the expanded scale. Image total lateral size = 10 nm. Line profiles of the normalized Electric Displacement Field $D_{fullscale} = 1$ are superposed onto the image and demonstrate the concentration effect of the water meniscus.

spot size of 600 nm. The samples were then transferred to the sMIM system and did not require any significant tip cleaning or conditioning for imaging.

**Scanning microwave impedance microscopy and scanning tunneling microscopy.** The AFM used to support the sMIM acquisition was an MFP-3D-SA manufactured by Asylum Research. The shielded co-axial AFM probes, as well as the electronics unit (model Scanwave Pro) used to transmit and measure the microwave signal was manufactured by PrimeNano Inc. The experiment schematics are shown in supplementary Fig. 1 in Supplementary Note 1: Experimental setup. All sMIM and AFM data were collected under ambient conditions. The phase calibration protocol was performed initially with cali- bration standards, as provided by the manufacturer. We later utilized the tip approach as a more convenient method to adjust the phase as during the tip approach, only the capacitance signal changes. We performed FEM simulations to verify that the real part of the reflected microwave signal did not change. When the real part of the reflected microwave signal no longer depended on the tip height $z$, the phase calibration process was completed. We performed several phase calibrations during an experimental set to verify electronics drift, tip contamination, or other factors that could impact the correct assessment of the measured impedance.

Scanning tunneling microscopy (STM): STM data were collected with a UHV VT STM/AFM model manufactured by Omicron GmbH. The tips used were etched tungsten probes. The STM was calibrated with a standard graphite lattice observed on a Highly Oriented Pyrolytic Graphite (HOPG) surface and a Si (111) reconstructed 7 × 7 surface lattice. All STM data were collected at room temperature at a pressure of $1 \times 10^{-10}$ Torr. The moiré patterns observed by STM were obtained from TBG samples initially supported by oxide-coated Si coupons for sMIM inspection. The bilayers were then, transferred to conducting, gold- coated mica coupons for subsequent STM analysis.

## Data availability
The data that support the findings of this study are available from the corresponding author upon reasonable request.

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

## Acknowledgements
The authors acknowledge financial support from CNPq, FINEP, FAPEMIG, INCT Nanomateriais de Carbono, and CAPES.

## Author contributions
Sample preparation: A.C.G., D.M., F.C.S., E.G.S.N. and L.C.C.; K.W. and T.T. provide hBN crystals; micro-Raman and TERS measurements: A.C.G. and E.G.S.N.; scanning probe microscopy measurements: D.A.A.O. and G.M.-R.; finite element computation: D. T., J.C.R., and C.G.d.R.; microwave technical support: W.A. and G.M.-R.; data analysis: G.M.-R. and D.A.A.O.; project idealization and guidance: A.J., J.C.R., C. G.d.R., and G.M. R.; paper writing: D.A.A.O. and G.M.-R. Some authors contributed with parts of the text and figures, and they all read and agreed on the final version of the manuscript.

## Competing interests
The authors declare no competing interests.
