## [Peer Review File · Nature Communications]

REVIEWER COMMENTS

Reviewer #1 (Remarks to the Author):

The authors present a very interesting advance in the field of near field microscopy, advocating the presence of water in the ambient being responsible for sMIM resolution better than 1 nm. The findings are novel and can open up many opportunities in nanoscale characterization. The manuscript is an "offspring" of another paper focused on the physics of low-angle tBLG, as revealed by a range of near field methods. While it is clear what the message of the current paper is and the experimental findings are nicely presented, the modelling and discussion parts are quite difficult to read with many seemingly random jumps to the supplementary. This is probably the result of the length limits, but the overall intelligibility of the paper suffers a lot. In my opinion, the authors should try to rewrite the mentioned sections. For example:

- 1) It is unclear what is the meniscus thickness and diameter, even with the aid of Fig S2 (a sketch depicting the basic parameters should be included in the main text anyway). Thickness probably means the z tip-sample separation (as it is 'defined' a few lines after it is first discussed), but I am not sure if this includes the thickness of the water layer on both or not.
- 2) Figure 2 a and b seem to contain some different data on the modelled menisci with different radii (or thickness, I am not sure): in a) the blue data points go to $z=2$ nm, in b) to 10 nm. a) contains analytical model (which is, by the way, not used anymore in the discussion), b) doesn't, but contains 1 nm meniscus.
- 3) Figure 2 caption seems to start somewhere in the middle.
- 4) The sentence "The model captures the capacitance dependence on z for most of the experimental range, except at the onset of meniscus formation." (page 8, line 157-158) is unclear: which model? Most of the experimental range is covered by the analytical model, and the rest by the FEM (?), or does it mean the FEM model with no meniscus is fine until the meniscus forms, then the meniscus-containing model has to be used? What happens when the model contains the meniscus with higher z? Or it does not form in this model?
- 5) It is unclear why the note on the absence of image contrast is in the main text – that would fit much better in supplementary. If the note has a specific importance, it should be clarified.
- 6) The meaning of the paragraph mentioning PSF (page 10, lines 196-201) is also not very clear, again not even with the aid of supplementary. Should be rewritten to contain some specific information in the main text and, if needed, an understandable discussion in the supplementary.

A minor point:

- 7) Please, make sure the variables/constants are defined (especially concerns the Introduction).

Reviewer #2 (Remarks to the Author):

This paper shown develops an interesting, potentially very impactful strategy to image 2D materials twist induced Moire structures at nanoscale resolution.

I believe this work would be of great interest to the community and should be published after modest revision.

The work builds on work by Seabron et al. (Their reference 11) who showed high resolution sMIM imaging of the quantum capacitance of carbon nanotubes. The authors should reference earlier work of Seabron, which illustrates the rationale for using a dielectric overlayer to optimize MIM imaging of quantum materials (Scanning Probe Microwave Reflectivity of Aligned Single-Walled Carbon Nanotubes: Imaging of Electronic Structure and Quantum Behavior at the Nanoscale: Seabron, E (Seabron, Eric); MacLaren, S (MacLaren, Scott); Xie, X (Xie, Xu); Rotkin, SV (Rotkin, Slava V.); Rogers, JA (Rogers, John A.); Wilson, WL (Wilson, William L.); ACS NANO Volume: 10, Issue: 1 Pages: 360-368; DOI: 10.1021/acsnano.5b04975 ; Published: JAN 2016.) This would provide better context for their observations.

Initially, how the sMIM images were collected is not clear. Were they capacitive grating maps? Were they just contact mode images? It's not clear from the text. The water meniscus serves as a defacto dielectric overlayer which enhances the capacitive response detected for the twisted bilayer, (correlating with the local conductivity map of the structure). The author should take advantage of the fact that they have both sMIM and UHV-STM to try to quantify the sMIM response. The MIM capacitive response is highly nonlinear. The described technique shows excellent contrast, but the signals are not quantitative. Correlation with STM would allow the researchers to try to quantify the reactance of the domain walls to get a feel for the carrier density sensitivity of sMIM.

Moreover, a simple study of the NAP dynamics as a function of relative humidity could also be informative. Humidity control should make the meniscus hydrodynamics more consistent impacting reproducibility.

We also note that there is an arXiv paper (arXiv:2006.04000), recently uploaded of a similar measurement with an alternative, (albeit unconvincing), argument for the high resolution observed.

Other issues:

1. Fig 1e has too low quality. It is hard to observe any moire substructure.
2. Fig. 3 is confusing. They claimed that water meniscus makes the difference. But in 3c it is not marked where the water meniscus is. It is also confusing that the water layer on both tip side and graphene side has no influence on the field profile if you look at 3a. But in 3c it shows confinement due to the water meniscus. Clarity here is needed.
3. Overall, I think their theory and the simulation needs to better link to the conductivity at the domain wall. The water meniscus may increase the contrast, the spatial resolution is set by the field profiles.

Reviewer #3 (Remarks to the Author):

Ohlberg et al. report on the experimental and simulation results of the immersion scanning microwave microscopy on twisted bilayer graphene. They use the near field microwave microscopy to image the moire superlattice in twisted bilayer graphene at various twisting angles. The authors have also provided COMSOL simulations to support and explain the high spatial resolution of their data and the simulations support their interpretation on the high resolution. The experimental results are interesting and useful. However, the current manuscript seems to fit a more specific or technical journal rather than Nature Communications.

First, the presentation quality of the current manuscript does not meet the bar of Nature Communications. The manuscript seems to be prepared in a rush. There are lots of unpolished phrases, undefined abbreviations, and inappropriate references. The authors mentioned in the abstract and include an independent section on tip enhanced Raman spectroscopy in the methods, but there are no Raman data shown in the manuscript. This is so unfortunate because the microwave near field microscope technique is very interesting, but the authors need better and

more professional presentations on their results.

Second, the scientific content in the current manuscript is quite thin. It sounds to me the current manuscript is more likely a technical description on the high-resolution of microwave near field microscope but not a high-profile paper in Nature series. The experimental data were only shown in figure 1 without further description on what phenomena they have observed and what are the interpretation and how to understand their observations on the interesting twisted bilayer graphene systems. Compared with a related manuscript (Ref. 12), the current manuscript contains significantly less experimental data, almost no analysis on the observation but only simulations on the technical advantage of the microwave near field microscope. Therefore, the current manuscript should better fit a more specific or technical journal rather than Nature Communications.

REVIEWER COMMENTS

Reviewer #1 (Remarks to the Author):

The authors present a very interesting advance in the field of near field microscopy, advocating the presence of water in the ambient being responsible for sMIM resolution better than 1 nm. The findings are novel and can open up many opportunities in nanoscale characterization. The manuscript is an “offspring” of another paper focused on the physics of low-angle tBLG, as revealed by a range of near field methods. While it is clear what the message of the current paper is and the experimental findings are nicely presented, the modelling and discussion parts are quite difficult to read with many seemingly random jumps to the supplementary. This is probably the result of the length limits, but the overall intelligibility of the paper suffers a lot. In my opinion, the authors should try to rewrite the mentioned sections. For example:

1) It is unclear what is the meniscus thickness and diameter, even with the aid of Fig S2 (a sketch depicting the basic parameters should be included in the main text anyway). Thickness probably means the z tip-sample separation (as it is ‘defined’ a few lines after it is first discussed), but I am not sure if this includes the thickness of the water layer on both or not.

2) Figure 2 a and b seem to contain some different data on the modelled menisci with different radii (or thickness, I am not sure): in a) the blue data points go to $z=2$ nm, in b) to 10 nm. a) contains analytical model (which is, by the way, not used anymore in the discussion), b) doesn’t, but contains 1 nm meniscus.

3) Figure 2 caption seems to start somewhere in the middle.

4) The sentence “The model captures the capacitance dependence on z for most of the experimental range, except at the onset of meniscus formation.” (page 8, line 157-158) is unclear: which model? Most of the experimental range is covered by the analytical model, and the rest by the FEM (?), or does it mean the FEM model with no meniscus is fine until the meniscus forms, then the meniscus-containing model has to be used? What happens when the model contains the meniscus with higher z? Or it does not form in this model?

5) It is unclear why the note on the absence of image contrast is in the main text – that would fit much better in supplementary. If the note has a specific importance, it should be clarified.

6) The meaning of the paragraph mentioning PSF (page 10, lines 196-201) is also not very clear, again not even with the aid of supplementary. Should be rewritten to contain some specific information in the main text and, if needed, an understandable discussion in the supplementary.

A minor point:

7) Please, make sure the variables/constants are defined (especially concerns the Introduction).

First and foremost, we thank the referee his/her generosity in dedicating time to provide a pathway to streamline the key message of the paper. We completely concur that constraints placed challenges in conveying our results in a clear fashion. We also agree that the information flow was hindered by excessive jumps to the supplemental material section. We discuss each point below:

- 1) It is unclear what is the meniscus thickness and diameter, even with the aid of Fig S2 (a sketch depicting the basic parameters should be included in the main text anyway). Thickness probably means the z tip-sample separation (as it is ‘defined’ a few lines after it is first discussed), but I am not sure if this includes the thickness of the water layer on both or not.

Indeed these parameters were not clearly defined originally. We added relevant info to the main text to fully describe the simulated geometry, referring to a new, revised fig.2 (see fig.2a., new). In the figure caption we added additional information to help the reader. We also improved fig.S4 adding key parameters (tip height z, thickness of the water layer on the tip, substrate, meshing strategy, and boundary conditions for calculating the system impedance).
Lines 407-431 of the supplemental material section.

- 2) Figure 2 a and b seem to contain some different data on the modelled menisci with different radii (or thickness, I am not sure): in a) the blue data points go to $z=2$ nm, in b) to 10 nm. a) contains analytical model (which is, by the way, not used anymore in the discussion), b) doesn’t, but contains 1 nm meniscus.

See previous comment. We modified the figure, caption and text. As for the analytical model, its purpose is to capture the essential dependence of the tip capacitance on z, and to validate the FEM

model. This way we have, for over two orders of magnitude, agreement between data, analytical model and FEM model, which gives us confidence to properly postulate the meniscus capacitance contribution to the paper discussion. We re-organized the narrative, making these points flow more freely.

- 3) Figure 2 caption seems to start somewhere in the middle.

The figure 2 caption in the submitted version was indeed poorly written. We completely modified it. Please see updated version.

- 4) The sentence “The model captures the capacitance dependence on z for most of the experimental range, except at the onset of meniscus formation.” (page 8, line 157-158) is unclear: which model? Most of the experimental range is covered by the analytical model, and the rest by the FEM (?), or does it mean the FEM model with no meniscus is fine until the meniscus forms, then the meniscus-containing model has to be used? What happens when the model contains the meniscus with higher z? Or it does not form in this model?

We rewrote the entire section, and organized the thoughts as follows:

Lines 138-146: discussion on the analytical model.

Lines 147-154: justification and rationale for using FEM model.

Lines 155-157: Water existence in all films and its key influence for $f=3\text{GHz}$

Lines 163-174: discussion on the FEM model results.

- 5) It is unclear why the note on the absence of image contrast is in the main text – that would fit much better in supplementary. If the note has a specific importance, it should be clarified.

The discussion was moved to the supplemental material next to experiment discussion, which was revised as per referee 2's request to include additional data pertaining to meniscus dynamics. We performed a new experiment that in the same scan clearly shows that upon meniscus rupture, no more imaging is possible (fig. S8).

- 6) The meaning of the paragraph mentioning PSF (page 10, lines 196-201) is also not very clear, again not even with the aid of supplementary. Should be rewritten to contain some specific information in the main text and, if needed, an understandable discussion in the supplementary.

Agreed. We removed the discussion on Point Spread Function (PSF). We did not have a good PSF to begin with, as it was 5nm in size. The moiré lattice as a lower bound for our resolution is already a very good and statistically significant measure.

A minor point:

- 7) Please, make sure the variables/constants are defined (especially concerns the Introduction).

Defined:

Lambda: **line 50**

Numerical aperture: **lines 50, 51**

Epsilon, epsilon_0, epsilon_r: **lines 79, 80**

Reviewer #2 (Remarks to the Author):

This paper shown develops an interesting, potentially very impactful strategy to image 2D materials twist induced Moire structures at nanoscale resolution.

I believe this work would be of great interest to the community and should be published after modest revision.

The work builds on work by Seabron et al. (Their reference 11) who showed high resolution sMIM imaging of the quantum capacitance of carbon nanotubes. The authors should reference earlier work of Sebron, which illustrates the rationale for using a dielectric overlayer **to optimize MIM imaging of quantum materials** (Scanning Probe Microwave Reflectivity of Aligned Single-Walled Carbon Nanotubes: Imaging of Electronic Structure and Quantum Behavior at the Nanoscale: Seabron, E (Seabron, Eric); MacLaren, S (MacLaren, Scott); Xie, X (Xie, Xu); Rotkin, SV (Rotkin, Slava V.); Rogers, JA (Rogers, John A.); Wilson, WL (Wilson, William L.); ACS NANO Volume: 10, Issue: 1 Pages: 360-368; DOI: 10.1021/acsnano.5b04975 ; Published: JAN 2016.) This would provide better context for their observations.

Indeed. We became familiar with the reference after going deeper into Seabron's PhD thesis and slides, and we concur that using hi-k dielectric contributes to this discussion and establishes a better context for the discussion we engage. We changed the text accordingly. Please see lines 74-76 in the revised text.

Initially, how the sMIM images were collected is not clear. Were they capacitive grating maps? Were they just contact mode images? (1) It's not clear from the text. The water meniscus serves as a defacto dielectric overlayer which enhances the capacitive response detected for the twisted bilayer, (correlating with the local conductivity map of the structure). The author should take advantage of the fact that they have both sMIM and UHV-STM to try to quantify the sMIM response. (2) The MIM capacitive response is highly nonlinear. (3) The described technique shows excellent contrast, but the signals are not quantitative. Correlation with STM would allow the researchers to try to quantify the reactance of the domain walls to get a feel for the carrier density sensitivity of sMIM. (4)

The referee brings a series of excellent points, which we address below (see numbers above for proper reference):

1) We modified the methods section explaining the sMIM measurements and phase calibration procedure. **Lines 246-253.**

2) We agree with the referee that establishing a direct correlation with STM experiments would improve the understanding of sMIM signal. There are a few issues worth mentioning for this particular comparison we put forth in fig. S3. a) We perform sMIM and STM experiments on the same TBG system, placing the TBG flake on both insulating substrate and on Au:Mica. We perform sMIM before and after the STM experiments. Performing the experiments on TBG:Au:Mica resulted in significant increase in reflection of the microwave signal, so we had to lower the incident power by 10dB. The real and imaginary components of the reflected microwaves stayed about the same ratio, so Au:Mica, or SiO₂ did not change the phase of the reflected signal, but substantially changed the intensity. b) STM results usually when performed in dI/dV mode, can bring the spectroscopy and LDOS parameters in a qualitative fashion. We did not perform spectroscopy, and therefore a direct comparison is not possible. What is possible and we could pursue eventually is to monitor dI/dV close to the Fermi level. This could give us a qualitative comparison between sMIM and STM. c) albeit the idea of correlating STM and sMIM is good and intriguing, we would need additional theoretical results such as Density Functional Theory calculations to link the LDOS, TBG conductivity and permittivity. We think a more complete work including theory will be a more satisfactory answer, but as is, the current work does not require these important assessments to validate our measured resolution (please see point 3 below, and figure S6 and relevant discussion).

3) We concur. Fig. S5 in the supplemental material shows the non-linearity in both real and imaginary parts of the reflected microwaves, as a function of the conductivity and permittivity of the surface.

4) As previously mentioned, the correlation with STM would require both spectroscopy data a DFT modelling. Although intriguing, we find that an in-depth discussion of the contrast mechanisms and correlation with STM and DFT would escape from our main theme, which is to explain the improved resolution. We shall defer this study for another publication, primarily because of space restrictions.

Moreover, a simple study of the NAP dynamics as a function of relative humidity could also be informative. Humidity control should make the meniscus hydrodynamics more consistent impacting reproducibility.

Nap experiments to probe meniscus dynamics can indeed be instrumental for a thorough understanding of the imaging process. According to Szoszkiewicz, the chief parameters governing meniscus nucleation are humidity, temperature, normal force and tip velocity. Changing the local temperature and humidity produces changes in the curvature radius r_k of the meniscus according to:

$$\frac{1}{r_k} = \left[\frac{k_B T}{\gamma_{LV} v_M} \right] \ln \left(\frac{p}{p_s} \right)$$

With T , k_B , γ_{LV} , v_M , and p/p_s as temperature, Boltzmann's constant, liquid-vapor water surface tension, molecular density of water, and relative humidity. Additionally, the nucleation time of a meniscus depends as well on these parameters, and also on the normal force applied to the cantilever.

In order to precisely change the humidity and temperature, one needs a significant amount of time in order to have stable operations. Szoszkiewicz and Choe in beautiful experiments covered most of the parameters, so in order to significantly advance our understanding from our experiments and this prior work, we chose to test meniscus stability by varying the nap lift height (which in essence is the force normal), and tip velocity, for constant humidity (51%) and temperature ($T=293K$).

Following the referee's suggestions and adapting to our current experimental constraints, we were able to define a very wide window of nap lifts (normal force) and tip velocity. The meniscus ruptures for the above conditions for a 300nm nap lift and 2um/s tip velocity. 300nm nap lift corresponds to about 6-8nm meniscus elongation, as the

majority of the deformation takes place at the cantilever. New data is presented in the nap section of the supplemental material, fig. s8. We added discussion addressing the points raised by the referee explaining fig. S8 in **lines 508-532** in the supplemental material.

In summary, we obtain great results and stable operation in a 40-60% humidity conditions and $T \sim 293\text{K}$, for operation in the attractive regime. The higher the lift, the slower the top tip velocity. For a substantial lift of 150nm, we were able to scan at all practical scan rates without meniscus rupture. Our findings indicate that as long as the tip is lifted, we obtain reproducible and relatively stable operation.

We also note that there is an arXiv paper (arXiv:2006.04000), recently uploaded of a similar measurement with an alternative, (albeit unconvincing), argument for the high resolution observed.

Indeed. We do cite this reference (ref. 13 in the updated manuscript, ref. 12 in the original manuscript), and have since updated as they were able to get it published. We have modeled their proposed geometry and also agree that it fails to describe our 1nm resolution data. The displacement vector footprint is too big to resolve their fine features even if one postulates the existence of a stable 5nm asperity such as the one they proposed.

Other issues:

1. Fig 1e has too low quality. It is hard to observe any moire substructure.

Fixed to a sample of higher twist angle of 4.5° , leading to a period of 3nm and resolution of 1.5nm (as an upper bound as determined by the Nyquist limit).

2. Fig. 3 is confusing. They claimed that water meniscus makes the difference. But in 3c it is not marked where the water meniscus is. It is also confusing that the water layer on both tip side and graphene side has no influence on the field profile if you look at 3a. But in 3c it shows confinement due to the water meniscus. Clarity here is needed.

Fig 3d is a close up at the tip apex of fig3c. Indeed, the goal of fig 3c is to show that the field is concentrated immediately at the meniscus, which is shown in more detail in fig3d. The water layer on both tip and substrate in the presence of the meniscus still has some displacement field, but note that the color scale for the data with meniscus is 80 times bigger. The caption was fixed to highlight the size of the simulation shown. Finally, the water layer on both tip and sample do not influence the field profile, as the referee noted. It only adds a shift in the calculated capacitance. We chose not to discuss the effect of water on both surfaces because it did not impact the resolution, and would only add a contribution to C_{stray} .

3. Overall, I think their theory and the simulation needs to better link to the conductivity at the domain wall. The water meniscus may increase the contrast, the spatial resolution is set by the field profiles.

This is an interesting suggestion. We verified what the displacement field footprint (in essence, the system resolution) on the surface immediately below the meniscus would be. We chose three different scenarios that would mimic an AA site (see figures S6a and S6e) of 5nm radius surrounded by AB stacking. The model parameter chosen to mimic this geometry was the conductivity, which we followed according to refs. S3 and S4. To look at the domain boundary would be very interesting, but because of the lack of axial symmetry and tremendous increase in computation time we chose to compare the AA site to the AB (see figures S6b and S6f) site and a system with only a hBN layer (see figures S6c and S6g). These conditions should cover most of the investigated cases and geometries. We then compare the extent of the \mathbf{D} field at the surface water layer, in order to see whether the substrate electronic properties would change the spatial resolution. Fig. S6.d shows a comparison between the three different profiles, and the key message is that despite the fact that the hBN layer shows a different profile (it peaks at 2nm, or twice the meniscus radius, producing a torus), that albeit of lower intensity, still retains the same order of magnitude resolution. In summary, as the referee suspected, even for surfaces of conductivities varying by orders of magnitude, the water meniscus sets the spatial resolution.
Lines: 448 – 470.

We thank referee 2 for the discussions, and although for a couple of the issues pointed out we would not be able to properly accommodate in the current format, we were able to address the major points and believe that the paper improved tremendously.

Reviewer #3 (Remarks to the Author):

Ohlberg et al. report on the experimental and simulation results of the immersion scanning microwave

microscopy on twisted bilayer graphene. They use the near field microwave microscopy to image the moiré superlattice in twisted bilayer graphene at various twisting angles. The authors have also provided COMSOL simulations to support and explain the high spatial resolution of their data and the simulations support their interpretation on the high resolution. The experimental results are interesting and useful. However, the current manuscript seems to fit a more specific or technical journal rather than Nature Communications. First, the presentation quality of the current manuscript does not meet the bar of Nature Communications. The manuscript seems to be prepared in a rush. There are lots of unpolished phrases, undefined abbreviations, and inappropriate references.

We are delighted that the referee finds the experimental results interesting and useful. We appreciate all the constructive criticism, that we reply below:

- 1) Unpolished phrases: we proofread the entire MS, and changed the language accordingly. As there were significant changes to accommodate requests from the other referees, a marked-up version was not helpful in verifying the changes.
- 2) As for the abbreviations, we corrected UHV and hBN (lines 102 and 130); HOPG in materials and methods section; SPM in authors contributions; RMS in the supplemental materials. We did not find any other undefined abbreviations.
- 3) NAP is not an acronym, so it was corrected. “Nap mode” stems from an aviation jargon: Nap-of-the-earth (NOE) is a type of very low-altitude flight course used by military aircraft to avoid enemy detection. This was already explained in the supplemental material, and the all-Caps was corrected.

The authors mentioned in the abstract and include an independent section on tip enhanced Raman spectroscopy in the methods, but there are no Raman data shown in the manuscript. This is so unfortunate because the microwave near field microscope technique is very interesting, but the authors need better and more professional presentations on their results.

There are two main issues concerning this request:

- 1) We accommodate the request by adding Raman data. We show in figure S2 in the newly added supplemental material S2 the confocal Raman microscopy of one entire flake. Using Raman as a guide allows us to land in regions that we want to investigate, as opposed to landing the tip in a random place. We discuss the adopted survey protocol. We also modified several figures to comply with a more professional presentation (Figs. 1, 2, S1, S2, S4, S6 and S8).
- 2) Our work on TERS is available as we point out in reference [arXiv:2006.09482](https://arxiv.org/abs/2006.09482), and correlations with sMIM can be done outside the scope of the present paper. Our goal here was to explain the unprecedented resolution, and a TERS x sMIM discussion would direct the readership to an important point, but not essential for this manuscript. Citing our reference therefore accomplishes the referee request.

Second, the scientific content in the current manuscript is quite thin. It sounds to me the current manuscript is more likely a technical description on the high-resolution of microwave near field microscope but not a high-profile paper in Nature series. The experimental data were only shown in figure 1 without further description on what phenomena they have observed and what are the interpretation and how to understand their observations on the interesting twisted bilayer graphene systems. Compared with a related manuscript (Ref. 12), the current manuscript contains significantly less experimental data, almost no analysis on the observation but only simulations on the technical advantage of the microwave near field microscope. Therefore, the current manuscript should better fit a more specific or technical journal rather than Nature Communications.

We do appreciate his/her efforts in improving the quality of our manuscript. Yet, we vehemently dispute the assertion “thin science”. Our results were derived from the realization that beating Abbe’s limit 10^8 was a result of the unique combination of near field operation and liquid immersion in an extremely high refractive index liquid, the ubiquitous water and its fortuitous nanometer scale meniscus and refractive index of 9 (at 3GHz). Previously, Seabron’s work was the only reference that acknowledged the importance of water. Not only our work addresses a required rigorous analysis of any sMIM data on Twisted Bilayer Graphene, its impact should reach a much wider audience as the concepts presented can be employed in a variety of other nanoscale tools and on the investigation of other nanosystems.

As a final note, the referee claims there is experimental data only in Fig.1. The referee should take note that in addition to the experimental data shown in fig. 1, figures 2, S2, S3, S7 and S8 have extensive novel experimental data with rigorous description and modelling of the observed phenomena.

REVIEWERS' COMMENTS

Reviewer #1 (Remarks to the Author):

The comments have been addressed and the manuscript's intelligibility has been much improved as a consequence. I have no further objections against its acceptance.

Reviewer #2 (Remarks to the Author):

I believe the authors have adequately addressed my concerns with the manuscript and I believe it is ready for publication.

Reviewer #3 (Remarks to the Author):

I thank the authors for their efforts to improve the manuscript. I will leave it to the editor to decide what to do with the revised manuscript regarding my concerns in the previous review report. Anyway, a careful polishing may be necessary for the current manuscript.